# Sirtuins as Metabolic Regulators of Immune Cells Phenotype and Function

**DOI:** 10.3390/genes12111698

**Published:** 2021-10-26

**Authors:** Lídia Fortuny, Carlos Sebastián

**Affiliations:** 1Department of Cell Biology, Physiology and Immunology, School of Biology, University of Barcelona, 08028 Barcelona, Spain; lfortuny@ub.edu; 2Institute of Biomedicine of the University of Barcelona (IBUB), 08028 Barcelona, Spain

**Keywords:** sirtuins, metabolism, immune cells, immune-related diseases

## Abstract

Beyond its role on the conversion of nutrients into energy and biomass, cellular metabolism is actively involved in the control of many physiological processes. Among these, it is becoming increasingly evident that specific metabolic pathways are associated with the phenotype of several immune cell types and, importantly, are crucial in controlling their differentiation, proliferation, and effector functions, thus shaping the immune response against pathogens and tumors. In this context, data generated over the last decade have uncovered mammalian sirtuins as important regulators of cellular metabolism, immune cell function, and cancer. Here, we summarize our current knowledge on the roles of this family of protein deacylases on the metabolic control of immune cells and their implications on immune-related diseases and cancer.

## 1. Introduction

Sirtuins are a family of highly conserved enzymes, homologues of the silencing information regulator 2 (Sir2) in yeasts, which was first identified as promoting lifespan and mediating gene silencing in Saccharomyces cerevisiae [1,2]. Sirtuins are found in organisms ranging from bacteria to humans. To date, seven isoforms (SIRT1–7) have been described in mammals that vary in their subcellular localization, enzymatic activity, molecular targets, and substrate specificity. SIRT1 and SIRT2 can be found in the cytoplasm and in the nucleus; SIRT3, SIRT4, and SIRT5 are mitochondrial sirtuins, while SIRT6 and SIRT7 have been reported to be found mainly in the nucleus [3]. However, the subcellular localization of these proteins is cell type dependent and can be regulated by environmental conditions [2,4]. For instance, SIRT1 is predominantly expressed in the cytoplasm in neurons, while spermatocytes express it only in the nucleus [5]. Moreover, its nucleocytoplasmic shuttling in cardiomyocytes is dependent on cell differentiation [5]. SIRT2 has also been shown to shuttle between the cytoplasm and the nucleus during mitosis [6], whereas some studies have shown that SIRT3 is mainly a nuclear protein that can be translocated to the mitochondria upon stress [7].

Sirtuins were first described as class III histone deacetylases following the observation that Sir2 was able to deacetylate histone H3 and H4 in yeasts [8]. However, it was later shown that these proteins can remove acetyl and other acyl groups (such as malonyl, succinyl, glutaryl, and long-chain fatty acyl groups) from both histone and non-histone proteins [2]. Furthermore, SIRT4 and SIRT6 function as ADP-ribosyltransferases [9,10,11]. One of the main features of this family of proteins is that, while other histone deacetylases are Zn^2+^ dependent, sirtuins require nicotinamide adenine dinucleotide (NAD^+^) to function. This reliance on NAD^+^ has placed sirtuins as metabolic sensors that translate the metabolic status of the cell to posttranslational protein modifications controlling cellular responses to metabolic stresses, including the regulation of transcription factors such as SREBPs, HIF1α, FoxO1, and PPARα, which regulate lipid and cholesterol homeostasis, glucose metabolism, gluconeogenesis, and β-oxidation, respectively [12,13,14,15].

A steadily growing body of work has established a link between metabolism and the immune system. The main function of the immune system is the elimination of infectious organisms and external agents, but it is also involved in the maintenance of tissue homeostasis, and its defective function leads to a set of immune-mediated diseases. In order to carry out their functions, the immune system drives responses that involve immune cell activation, cytokine and chemokine secretion, and differentiation and proliferation of innate and adaptive immune cells. Work undertaken during the last decade has revealed that most of these processes are regulated by cellular metabolic pathways that are essential to generate appropriate immune responses. All this work has resulted in an emerging field called immunometabolism [16]. Importantly, this crosstalk between metabolism and the immune system could have a profound impact on the design of novel therapies controlling effector functions and reducing pathological inflammation in several diseases, such as autoimmune diseases and cancer [17]. In the present review, we summarize our current view on the crosstalk between sirtuins and immunometabolism and the roles of this family of proteins in controlling the fate and function of innate and adaptive immune cells. Furthermore, we discuss whether sirtuins might be potential targets for novel therapies in immune-mediated diseases such as inflammation, autoimmune conditions, and cancer.

## 2. Sirtuins Translate Metabolic Cues to Immune Responses

Immune cells exhibit different proliferative, differentiation, and activation states, which are closely connected to specific metabolic requirements. Resting immune cells usually exhibit a catabolic metabolism to maintain homeostasis; thus they preferentially rely on oxidative phosphorylation (OXPHOS) and fatty acid oxidation. However, upon activation, most of the immune cells switch to an anabolic metabolism in order to generate precursors required for their proliferation and expansion [16,18]. These anabolic pathways are mainly supported via aerobic glycolysis, a metabolic reprogramming similar to the one observed in cancer cells and first described by Otto Warburg in the 1920s [19]. Cells that rely on the Warburg effect tend to increase glucose uptake, diverting it into anaplerotic pathways and preferentially convert glucose-derived pyruvate into lactate, even in the presence of oxygen. Although this pathway is less efficient in producing ATP compared to OXPHOS, it allows the fast generation of biosynthetic precursors necessary for immune cell proliferation. This switch to aerobic glycolysis has been reported under inflammatory conditions such as sepsis and lipopolysaccharide (LPS) stimulation, as well as in some autoimmune diseases such as rheumatoid arthritis [20,21]. Beyond its role in supporting cell proliferation, immune cell metabolism actively regulates the differentiation, polarization and activation of several immune cell types such as T cells and macrophages, as summarized in the next sections. Importantly, this crosstalk between metabolism and immune cell function is bidirectional, as activation and cytokine conditions directly impact immune cell metabolism.

Due to their roles as metabolic regulators, sirtuins could have a key role in translating metabolic cues into immune responses and actively participate in the regulation of immunometabolism. Indeed, accumulating evidence during the last few years have demonstrated that sirtuins have an important role in modulating inflammatory stress responses by regulating the bioenergetic metabolism of immune cells, as summarized in Table 1 [22,23,24]. Most of this work has focused on the role SIRT1 on T cell and macrophage metabolism, yet recent data have expanded our knowledge to other sirtuins and immune cell types. 

### 2.1. The Role of Sirtuins on Innate Immune Cells

Innate immunity provides the initial defense against infections and it develops quickly after antigen recognition, which orchestrates the action of granulocytes, monocytes/macrophages, dendritic cells (DCs) and natural killer cells (NK). Among these, it has been shown that sirtuins have an important role in regulating immunometabolism in macrophages and DCs [59].

#### 2.1.1. Macrophages

Macrophages are specialized immune cells involved in the phagocytosis and elimination of pathogens and death host cells. They are also responsible for antigen presentation to T cells and the production of inflammatory cytokines, thus coordinating an effective immune response. Macrophages originate from circulating blood monocytes that leave the circulation and migrate to different tissues, where they differentiate. Among all the immune cells, monocytes/macrophages are one of the most widely studied in the field of immunometabolism, and recent evidence shows that sirtuins may modulate their fate and function [60]. 

Phenotypically, macrophages are plastic cells that can be generally divided into pro-inflammatory M1 or anti-inflammatory M2 macrophages. These functional states depend on the stimuli that they receive and their following activation. M1 are classically activated through Toll-like receptors (TLRs) and release high levels of pro-inflammatory cytokines such as tumor necrosis factor-α (TNFα), interleukin-1 (IL-1), or IL-6. On the other hand, M2 macrophages are alternative activated by IL-4 or IL-13 and are involved in the resolution of the inflammation by the release of anti-inflammatory cytokines such as IL-10 and transforming growth factor β (TGFβ) [61]. Intriguingly, M1 and M2 macrophages differ not only in their inflammatory phenotypes but also in their metabolism. Pro-inflammatory M1 macrophages mainly rely on glycolysis through the activation of several transcription factors such as HIF1-α and exhibit an impaired tricarboxylic acid (TCA) cycle. On the contrary, anti-inflammatory M2 macrophages maintain an active TCA cycle to support OXPHOS for energy production. In addition, these two types of macrophages also differ in the way they metabolize amino acids and lipids. Under pro-inflammatory conditions, such as stimulation with LPS, TNF-α or IFN-γ, expression of nitric oxide synthase 2 (NOS2) drives the conversion of arginine to citrulline and nitric oxide (NO), a key mediator of the anti-microbial activity of macrophages. In contrast, M2 macrophages induce the expression of arginase 1 (*Arg1*), which catabolizes arginine to ornithine, a precursor of polyamines synthesis important for cell growth and tissue repair [61]. Glutamine is another amino acid differentially metabolized by macrophages. While M1 macrophages channel glutamine into the TCA cycle to promote succinate synthesis, leading to HIF1-α stabilization [62], M2 macrophages engage glutaminolysis to generate α-ketoglutarate, which regulates the M2 phenotype by acting at different levels, including OXPHOS and fatty acid oxidation (FAO), epigenetic reprogramming, inhibition of HIF1γ expression, and UDP-GlcNAc synthesis [63]. M2 macrophages are also able to synthesize glutamine from glutamate and ammonia through induction of glutamine synthase (GS), which is barely expressed in M1 macrophages [64]. Regarding lipid metabolism, it has been documented that M1 macrophages preferentially induce fatty acid synthesis (FAS) to support energy production and prostaglandin biosynthesis, whereas M2 macrophages rely on fatty acid uptake and oxidation, mainly driven by the coordinated action of STAT6, PPAR and PGC1 [61]. 

From all these studies, it is clear that macrophage polarization requires a metabolic reprogramming that may be supported by sirtuins. As mentioned above, activation of macrophages by LPS induces a metabolic switch towards aerobic glycolysis impairing mitochondrial metabolism, which results in changes in NAD^+^/NADH ratios, potentially affecting sirtuin activity. Indeed, in a model of sepsis, NAD^+^ levels have been shown to regulate a metabolic shift in macrophages from glycolytic metabolism to increased FAO by a mechanism involving SIRT1 and SIRT6 (Figure 1A) [25]. Increased glycolysis during the early inflammatory phase of sepsis results in high concentration of NAD^+^, which induces SIRT6 and SIRT1 activity. SIRT6, by acting as a corepressor of HIF1α activity [13], inhibits glucose metabolism, while SIRT1 promotes FAO through activation of PGC1α. Importantly, the combined action of these two sirtuins enables the bioenergetic shift required for the transition to the late adaptation to sepsis [25]. These results are in agreement with the fact that SIRT1 is highly expressed in M2 macrophages compared to M1 and its specific deletion in macrophages stimulates the infiltration of M1 macrophages and reduces the amount of M2 macrophages in adipose tissue [26]. In this context, it has been shown that IFNγ inhibits the expression of SIRT1 in skeletal muscle cells by upregulating the expression of *HIC1*, a well-known repressor of SIRT1 expression [65]. Although the presence of this regulatory axis in macrophages is not known, these results suggest that this pro-inflammatory cytokine could dampen SIRT1-driven oxidative metabolism favoring M1 polarization of macrophages.

Further evidence linking sirtuins with macrophage metabolism and function involve their crosstalk with several key signaling pathways in immune cells (Figure 1B). NF-κB is a transcription factor directly involved in the transcription of several genes induced by pro-inflammatory cytokines. Although NF-κB is best known for its role in immune responses and inflammation, growing evidence suggests its implication as a regulator of core cellular metabolic pathways, including glycolysis, regulation of triglyceride levels, lipogenesis, and fatty acid metabolism [66]. NF-κB has been shown to induce glycolysis by directly increasing the expression of the glucose transporter *GLUT3* [67] and to upregulate glycolytic metabolism during acute inflammation [68]. Remarkably, SIRT1 has been demonstrated to inhibit NF-κB signaling and to enforce oxidative metabolism associated with the resolution of inflammation. SIRT1 directly deacetylates the RelA subunit of NF-κB, also known as p65, at lysine 310 (K310) inhibiting its transcriptional activity [27]. Concomitantly, myeloid deletion of SIRT1 causes hyperacetylation of NF-κB in bone marrow-derived macrophages (BMDMs), resulting in an enhanced transcriptional activity and an increase in the secretion of the pro-inflammatory cytokines TNFα and IL-1β [31]. Consistent with increased pro-inflammatory gene expression, these mice also exhibit higher insulin levels and an impaired glucose metabolism associated with high levels of M1 activated macrophages in liver and adipose tissue. Furthermore, SIRT1-dependent oxidative metabolism is also driven by activation of AMPK, PPARα and PGC1α, which have been involved in the inhibition of NF-κB activity, thus suppressing inflammation [28,29,30]. However, a different study found that NF-κB acts as a regulator of mitochondrial respiration by suppressing the Warburg effect (glycolysis) via the upregulation of cytochrome c oxidase 2 (*SCO2*), therefore stimulating OXPHOS [69]. These apparently contradictory results seem to be dependent on the cell type studied and its p53 status [70].

Similar to SIRT1, other sirtuins have been described to regulate NF-κB activity as well (Figure 1B). SIRT2 deacetylates p65 subunit and inhibits pro-inflammatory gene expression in BMDMs [33,34]. However, another study reported a decrease in pro-inflammatory markers and lower NF-κB activation in BMDMs from SIRT2-deficient mice [35]. SIRT6 has also been associated with NF-κB inhibition by directly deacetylating lysine 9 (K9) of p65 and by physically interacting and stabilizing IκB [71,72]. Lastly, SIRT5 has been shown to promote acetylation of NF-κB by competing with SIRT2 for its binding to p65 [38]. Altogether, these studies suggest that sirtuins could have an important role in the regulation of macrophage metabolism and their inflammatory phenotype by controlling the activity of NF-κB.

Besides NF-κB, other transcription factors important for immune cell function modulate metabolism and are regulated by sirtuins (Figure 1B). Myelocytomatosis oncogene (Myc) regulates the transcription of many genes involved in immune responses and is a key factor governing the metabolic reprogramming of macrophages. Myc is associated with an M2 phenotype, and its expression increases glycolysis and glutaminolysis, both necessary for macrophage proliferation [73]. The transcription factor E2F1 has been shown to repress glucose oxidation and to promote glycolysis in several tissues. E2F1 knock-down results in an increased expression of mitochondrial biogenesis and OXPHOS-related genes [74]. Importantly, a recent study showed that Myc and E2F1 target genes were down-regulated upon SIRT1 inhibition, directly affecting macrophage self-renewal and revealing that sirtuins may also modulate macrophage metabolism reprogramming by regulating the activity of these transcription factors [32]. This same study also showed that, contrarily to Myc and E2F1, Forkhead box protein (FoxO) transcription factors were up-regulated following SIRT1 inhibition and that this sirtuin retained FoxO inactive in the cytoplasm [32]. In line with this, using multi-omics analysis, Yan et al. (2020) reported that deficiency of FoxO1 promotes the differentiation of M2 macrophages, then lowering their inflammatory phenotype as well as down-regulating glycolysis [75]. Similarly, SIRT3 overexpression stimulates M2 macrophage polarization, a phenotype associated with increased acetylation of FoxO1, which blocks its translocation to the nucleus [37]. 

Hypoxia-inducible factor-1 α (HIF-1α) is considered a master modulator of glucose metabolism. Stabilization of this transcription factor under hypoxia leads to the up-regulation of the expression of several glycolytic genes and the inhibition of OXPHOS by up-regulating the expression of pyruvate dehydrogenase kinase (*PDK*), thus inhibiting pyruvate oxidation in the mitochondria [76]. Several sirtuins have been involved in the regulation of HIF-1α. SIRT6 has been shown to act as a co-repressor of *HIF-1α* transcriptional activity by deacetylating H3K9 and H3K56 on the promoters of several glycolytic genes, thus repressing their expression. Consequently, SIRT6 loss leads to increased glycolytic gene expression, glucose uptake, and lactate production [13]. Remarkably, repression of *HIF-1α* by SIRT6 drives a metabolic shift from glycolysis to fatty acid oxidation during sepsis [25]. However, combined deletion of SIRT2 and SIRT3 in mice resulted in BMDMs more dependent on fatty acid oxidation and with an increased expression of *HIF-1α* [36]. Lastly, it has been shown that SIRT5 directly interacts with and desuccinylates PKM2, a coactivator of HIF-1α. Overexpression of SIRT5 decreases the pyruvate kinase activity of PKM2 and suppresses the pro-inflammatory response of macrophages [39]. Interestingly, these results are in contrast with the role of SIRT5 in promoting the innate inflammatory response in macrophages by regulating NF-κB activity [38], suggesting that more specific studies on the role of sirtuins on immune cells are needed in order to completely understand how they regulate their metabolism.

#### 2.1.2. Dendritic Cells 

Together with macrophages, DCs are in the frontline of innate immunity, and they are rapidly activated upon antigen recognition. The main function of these cells is to capture and process antigenic material and present it on the cell surface within the major histocompatibility complex class II (MHC class II) to naïve T cells, leading to their activation and initiating an adaptive immune response. Depending on the stimuli received, DCs release a different pattern of cytokines that will drive the differentiation of T helper (Th) cells to Th1, Th2, Th17, or T regulatory (Treg) cells [77]. Similar to macrophages, once activated, DCs undergo a shift in their metabolism from OXPHOS to Warburg-like metabolism [78]. It has been shown that, upon TLR activation with LPS, DCs switch their metabolism to glycolysis, which is critical for activation of T cells and the release of IL-6 and TNF-α [79]. Overall, DCs metabolic reprogramming is similar to the one observed in macrophages, and it may involve some of the same mechanisms. Therefore, as sirtuins are able to modulate macrophage immunometabolism, they could also be implicated in the regulation of DCs metabolism. However, as of today, little research has been done in the role of sirtuins specifically modulating DCs metabolism. 

Although T cells will be discussed later in the current review, an important note is that sirtuins have shown to regulate the fate of T cells through the regulation of DCs, and this could be associated with their roles on metabolic reprogramming. Legutko et al. (2011) reported that SIRT1 promotes Th2 responses through the inhibition of peroxisome proliferator-activated receptor-γ (PPARγ) activity, which regulates fatty acid storage and glucose metabolism [40]. PPARγ is a nuclear receptor that has recently emerged as an important factor in the maturation and function of some immune cells. PPARγ modulates the immune response through regulation of DCs function by controlling the expression of genes involved in lipid metabolism. Furthermore, it exerts an immunomodulatory role and represses inflammation by inhibiting NF-κB and AP-1 in macrophages [41]. Another study demonstrated a role for SIRT1 in antagonizing the acetylation of interferon regulatory factor 1 (IRF1) in DCs, thus inhibiting the expression of IL-27 and IFN-β. In line with this, SIRT1 deletion in DCs suppresses the differentiation of Th cells into Th17 [42]. Importantly, these mice exhibit an attenuated phenotype during experimental autoimmune encephalomyelitis (EAE), a mouse model of multiple sclerosis (MS), suggesting its potential role in modulating autoimmune diseases and inflammation [42]. Intriguingly, SIRT6 has been shown to be involved in EAE as well, as SIRT6 inhibition delayed the development of EAE directly through the impairment of DCs migration [45]. Moreover, SIRT6 is essential for bone marrow-derived dendritic cell (BMDC) generation and exerts different effects on cytokine production by these cells. According to Lasiglie et al. (2016), SIRT6 deletion in mice results in a reduced expression of TNF-α and IL-12 and an increased IL-6 production in baseline conditions. However, after TLR stimulation, BMDCs derived from SIRT6 KO mice secrete higher levels of TNF-α and IL-6, which can be associated with NF-κB overactivation [80]. These data suggest that the effects of sirtuins in modulating immune cells can vary depending on the type and the state of the cells and the immune response studied. Altogether, it is clear that sirtuins regulate the function of DCs and therefore influence Th polarization as well. However, more research is needed to address the potential role of sirtuins on DC metabolism and their mechanisms of action. 

In addition to modulating Th cell fate through DCs, sirtuins can also regulate transcription factors that play a key role on immunometabolism. As in the case of macrophages, sirtuins can regulate the transcription factor NF-κB in DCs. Gogoi et al. (2020) demonstrated that the mRNA levels of SIRT2 are upregulated after Salmonella infection in BMDCs, suggesting that SIRT2 has a critical role in regulating the immune response to the pathogen [44]. In addition, chemical inhibition of SIRT2 resulted in increased acetylation of the p65 subunit and inhibited the translocation of NF-κB to the nucleus. As discussed above, a SIRT2-NF-κB axis regulates macrophage metabolism, and thus, it is tempting to speculate that this transcription factor might also regulate immunometabolism in DCs. 

Another transcription factor that has been shown to be modulated by sirtuins in DCs is HIF-1α. As mentioned before, *HIF-1α* is key in upregulating glycolysis. Liu et al. (2015) reported that, similar to macrophages, SIRT1 deficiency in DC cells resulted in the accumulation of *HIF-1α* while SIRT1 stimulation decreased *HIF-1α* expression [43]. In addition, SIRT1 influenced the cytokine production of DCs through *HIF-1α*, including IL-12 and TGF-β1, thereby directing the differentiation of T cells.

Besides macrophages and DCs, the role of sirtuins on the metabolic control of other innate immune cells has not been described. However, it is known that neutrophils undergo a metabolic shift towards glycolysis to support their phagocytic activity [81]. Based on this, one can hypothesize that those sirtuins involved in the control of glycolysis could regulate this metabolic pathway in neutrophils as well.

### 2.2. The Role of Sirtuins on Adaptive Immune Cells

The adaptive immune response is characterized by the generation of specific antibodies and cellular responses that allow the efficient elimination of specific antigens. Importantly, a defining feature of adaptive immunity is the ability to produce memory cells, which will enhance the immune response during repeated infections. The main cell types that carry out this immune adaptive response are T and B lymphocytes. Similar to other immune cells, the metabolic features of T and B cells are optimized to support their functions, and recent data indicate that the tight regulation of metabolism in these cells could involve sirtuins in a similar way to the innate immune response [82].

#### 2.2.1. T Cells

T cells are produced from the lymphocytic lineage in the bone marrow and complete their maturation in the thymus to express CD4 or CD8 in their cell surface. Upon their activation, T cells proliferate and acquire effector functions in a very regulated process coordinated by cytokines and transcription factors. As in other cells of the immune system, T cell activation leads to a transition from a catabolic to an anabolic state necessary to fulfil T cell effector cellular needs, which, in part, is controlled by sirtuins [83]. Naïve T cells mainly rely on OXPHOS and fatty acid oxidation, while activated T cells increase nutrient uptake, aerobic glycolysis, and glutaminolysis and upregulate the expression of glucose and amino acid transporters to provide metabolic intermediates for nucleotide and fatty acid synthesis [83]. Depending on the stimuli received during their activation, both CD4+ and CD8+ T cells will differentiate into effector or memory T cells. These cell subsets will present unique features and functions to eliminate antigens and are supported by specific and optimized metabolic programs [82].

The best described subsets of CD4+ effector T cells are T helper 1 (Th1), Th2, Th9, and Th17. Each of these T cell types is specialized in generating a specific response against an infectious agent by promoting a characteristic cytokine profile and stimulating specific immune cells. Furthermore, CD4+ T cells can differentiate into regulatory T cells (Treg), which exhibit immunosuppressive functions and reduce inflammation. On the other hand, CD8+ T cells can be classified into effector T cells, which are responsible for removing infectious agents by releasing cytotoxic cytokines and molecules and memory CD8+ T cells [82]. This is a general classification, and many different subtypes can be generated depending on the stimuli that T cells receive during their maturation. 

CD4+ effector T cells have been shown to be highly glycolytic and to express high levels of GLUT1 in their cell surface. This metabolic rewiring is essential for the clonal expansion necessary for their functions, and it is regulated by the mTOR pathway. However, Treg cells rely on lipid oxidation and express low levels of GLUT1 [51,84]. Similarly, cytotoxic CD8+ T cells also rely on glycolysis and present high rates of glucose and amino acid uptake to support their cytokine production, but remarkably, these cells switch back to a catabolic metabolism when they become CD8+ memory T cells [52]. In addition, fatty acid metabolism has been shown to be key in regulating T cell polarization. Cytotoxic CD8+ T cells display increased fatty acid synthesis, while memory CD8+ T cells switch to the catabolism of fatty acids by activating fatty acid oxidation [51]. Similarly, Lochner et al. (2015) also reported that the development of Th1, Th2, and Th17 subsets depends on fatty acid synthesis, but this metabolic pathway is not essential for Treg differentiation [51]. Altogether, these results suggest a strong relationship between metabolism and differentiation and demonstrate that the regulation of immunometabolism is determinant for the fate of T cells. 

Sirtuins have emerged as critical regulators of T cell differentiation and function. SIRT1 expression is abundant in the thymus, suggesting a key role of this sirtuin in T cell development. Indeed, SIRT1 deficiency in mice is associated with an increased T cell activation and a breakdown of CD4+ T cell tolerance [46]. Accordingly, SIRT1-KO mice exhibit a more pro-inflammatory T cell phenotype and increased T cell proliferation and are more susceptible to develop autoimmune diseases. SIRT1 reduces the production of Th1 and Th2-associated cytokines in CD4+ T cells, likely through AP-1 inhibition [46]. However, other studies have reported that SIRT1 promotes Th2 differentiation by repressing PPARγ in DCs (as discussed above) [40]. Although these studies do not directly link metabolism with T cell differentiation, it is tempting to speculate that some of the effects of SIRT1 on Th cells differentiation might involve SIRT1-dependent metabolic reprogramming. Indeed, this is the case, for instance, of Th9 differentiation (Figure 2A). It has been shown that SIRT1-dependent glycolysis modulation is involved in Th9 cell differentiation through an mTOR-HIF1α-dependent pathway [47]. Genetic deletion of SIRT1 in CD4+ T cells or siRNA-mediated downregulation of SIRT1 expression in human T cells increase IL-9 production and glycolytic metabolism, while ectopic SIRT1 expression has the opposite phenotypes. Importantly, SIRT1-dependent control of Th9 cells differentiation appears to have a key role on anti-tumor immunity and allergic pulmonary inflammation [47], suggesting that targeting sirtuins could be exploited as potential therapies for inflammatory and autoimmune diseases.

Th17 cell differentiation is also modulated by SIRT1, yet it is unclear whether this sirtuin plays a stimulating or inhibitory role on the Th17 lineage. It has been described that SIRT1 promotes Th17 differentiation, therefore stimulating inflammation [48]. SIRT1 deletion in CD4 T cells or its pharmacological inhibition by nicotinamide suppresses Th17 differentiation and protects against autoimmune diseases. However, a different study reported that SIRT1 activation with NAD^+^ reduced inflammatory responses probably by suppressing Th1 and Th17 differentiation [49]. In the same line, Limagne et al. (2017) demonstrated that pharmacological activation of SIRT1 with resveratrol, metformin, and SRT1720 blocked Th17 cell differentiation [50]. According to this study, endogenous activation of SIRT1 may enhance Th17 responses via RORγt deacetylation, while a more potent pharmacological activation of SIRT1 could induce the inhibition of Th17 differentiation via STAT3 deacetylation. Indeed, RORγt is key in Th17 differentiation, which is associated with an increase in the expression of cholesterol biosynthetic genes, linking cholesterol metabolism with immune function [85]. STAT3 is also involved in cellular metabolism, and its activation promotes glycolysis and inhibits OXPHOS through a HIF-1α-dependent mechanism [86]. These studies suggest that sirtuins may modulate Th17 differentiation by directly modulating the metabolism of T cells. Although the effects of other sirtuins on Th17 cell differentiation have not been studied, other members of the family could be implicated in this metabolic reprogramming. For example, SIRT6 represses the transcription factor HIF-1α, and its deficiency upregulates glycolysis and diminishes OXPHOS, as discussed earlier [13]. Interestingly, *HIF-1α* enhances Th17 cell differentiation via RORγt activation, suggesting that regulation of this transcriptional factor by SIRT6 might also be involved in modulating the fate of T cells [87]. 

In contrast to other Th subtypes that are highly glycolytic, Treg metabolism relies on lipid oxidation to support their activity [82,84]. The transcription factor forkhead box P3 (*FoxP3*) has been reported to be essential for Treg differentiation and function. Importantly, *FoxP3* acetylation prevents its lysosomal degradation increasing *FoxP3* protein levels [88], potentially linking sirtuin activity with Treg differentiation. In this context, it has been reported that SIRT1 inhibition enhances Treg suppressive function and SIRT1 deletion in CD4 T cells promotes the expression of *FoxP3* in Treg cells [89]. Supporting these data, Daenthanasanmak et al. (2019) demonstrated that SIRT1 deletion in T cells leads to an increase in p53 acetylation, promoting Treg stability, thus making sirtuins a potential target for the treatment of certain autoimmune diseases and allograft survival [90]. Remarkably, similar results were obtained in human T cells treated with the SIRT1 inhibitor Ex-527 [90]. SIRT3 has been also linked to Treg modulation, but contrary to SIRT1, loss of SIRT3 in mice impairs Treg suppressive function, likely due to the role of SIRT3 in promoting oxidative metabolism (Figure 2A) [55]. 

Lastly, SIRT1 also regulates CD8+ memory T cells (Figure 2B). Terminally differentiated memory T cells have an enhanced capacity to use glycolysis, which is associated with a decreased expression of SIRT1. Mechanistically, decreased SIRT1 expression enhances proteasomal degradation of FoxO1 and promotes an increased glycolytic metabolism and granzyme B secretion, highlighting the role of the SIRT1-FoxO1 axis in regulating metabolism and function of resting memory T cells [53]. Remarkably, SIRT1 is also involved in the differentiation of CD8+ effector cells through epigenetic and metabolic reprogramming (Figure 2B). Transcriptional repression of *Sirt1* by the basic leucine zipper transcription factor ATF-like (BATF) during the late effector stage leads to increased histone acetylation of the T-bet locus and increased cellular NAD^+^, resulting in increased levels of ATP and effector T cells differentiation and survival [54]. Therefore, the function and fate of T cells might also be driven in part by sirtuins, similarly to other immune cell types. 

#### 2.2.2. B Cells

As with T cells, B cells belong to the lymphocytic linage and are produced in the bone marrow. Immature B cells undergo several gene rearrangements at immunoglobulin loci and finish their early development in the spleen to differentiate into follicular (FO) or marginal zone (MZ) B lymphocytes. Mature B cells can remain in the spleen or migrate to peripheral lymph nodes. Upon activation, B cells contribute to the humoral immunological response by secreting antibodies. Moreover, a subset of mature B cells will differentiate into memory cells, which will retain the ability to secrete high affinity antibodies during a second infection [91]. While in other cell types, such as macrophages or T cells, metabolism has been extensively studied, little research has been performed focusing on the immunometabolism of B cells and their changes between different cell states. 

In contrast to other immune cells that increase glucose uptake and glycolysis upon activation, B cells rather increase metabolism in a more general and balanced way. B cells not only upregulate glucose uptake and lactate production after LPS stimulation but also OXPHOS. Interestingly, this activation-induced metabolic reprogramming is affected during B cell tolerance, as anergic B cells failed to increase metabolism following activation, while chronically stimulated B cells from Systemic Lupus Erythematosus-like disease mice displayed a rapid induction of aerobic glycolysis [92]. Despite having a balanced metabolism, active glycolysis has been shown to be essential for B cell proliferation and antibody secretion [92,93], which, in part, is regulated by phosphatidylinositol 3-kinase (PI3K) activity [94]. Besides glycolysis, B cell proliferation and expansion also require an increased supply of lipids and cholesterol [95]. Therefore, as in other immune cells, metabolism plays a key role in regulating the function of B lymphocytes, and sirtuins might be an important element in this regulation. Although little is known about the effect of sirtuins in B cells, evidence suggests that SIRT1, SIRT2, SIRT3, and SIRT4 could modulate their metabolism [82,93]. However, it is important to note that most of these studies have been performed in pathological conditions and not in healthy lymphocytes, so these results may be slightly different in the latter [93]. 

SIRT1 deletion in mice has been directly associated with deposits of immune complexes in the liver and the kidneys [96]. Furthermore, the expression of miR-132, which suppresses SIRT1, is increased in MS patients [56]. These results suggest that sirtuins could be associated with different autoimmune diseases, and it has been proposed that B cell immunometabolism could have an important role in these conditions [93]. Sirtuins have been also studied in the context of B cell malignancies, such as chronic lymphocytic leukaemia (CLL), which is characterized by an increased abnormal production of B lymphocytes. It has been reported that the expression and function of SIRT1 and SIRT2 are upregulated in CLL cells, and their inhibition halts their proliferation [57]. Although this study did not focus on characterizing the metabolic phenotype of these cells, some evidence suggests that metabolic reprogramming could be key to the progression of CLL. In fact, loss of SIRT3 in human CLL cells contributes to cancer progression via hyperacetylation of the metabolic enzymes isocitrate dehydrogenase (IDH2) and superoxide dismutase 2 (SOD2). Furthermore, in the same study, the authors showed that SIRT3 overexpression suppressed Warburg-like metabolism and decreased glycolytic gene expression in CLL cells [58]. Taken together, these studies suggest a possible role for sirtuins in B cell malignancies, although this role may differ depending on the disease and sirtuin studied. 

Lastly, despite the lack of direct experimental evidence on the crosstalk between sirtuins and B cell metabolism, it is tempting to speculate that this family of protein deacylases could potentially regulate B cell metabolism through their functional interaction with transcription factors important for B cell function and metabolism. For instance, *HIF-1α* is key for the production of IL-10 by B lymphocytes, and its expression increases in activated B cells, controlling their expansion by regulating glycolytic metabolism [97]. Furthermore, the deficiency of this transcription factor exacerbates autoimmune diseases, such as EAE and arthritis [97]. However, another study showed that the metabolic reprogramming observed upon B cell activation is independent of *HIF-1α* and dependent on Myc [92]. Nevertheless, both *HIF-1α* and MYC are targets of several sirtuins, suggesting that sirtuins could be at the intersection of B cell function and metabolism.

## 3. Sirtuins at the Crossroad of Metabolism and Immune-Related Diseases

Sirtuins have long been considered possible therapeutic targets for several age-associated pathologies, including neurodegeneration. In light of the data summarized in this review, it is reasonable to conceive sirtuins as important players in some immune-related diseases as well [4]. In this section we discuss the role of sirtuins on immunometabolism imbalance and its involvement in inflammation, autoimmune diseases, and cancer.

As discussed above, SIRT1 and SIRT6 have a major role on the regulation of the acute inflammatory response during sepsis by controlling the transition from an early inflammatory response to a late stage where immune cells enter in a resting state and resolve the infection [24]. This transition is accompanied by an increase in NAD^+^ and the activation of SIRT1 and SIRT6, which leads to a shift from aerobic glycolysis to FAO (Figure 1A). Moreover, NAD^+^ is essential to restore OXPHOS and homeostasis in human monocyte-derived macrophages while pharmacological blockade of de novo NAD^+^ synthesis impairs the resolution of inflammation [98]. These results suggest that sirtuins, as NAD^+^ sensors, might also be involved in this process. Besides their role in sepsis, sirtuins may also be involved in other inflammatory-mediated conditions (as discussed below) and have been proposed as potential targets for novel therapies for the treatment of autoimmune diseases and leukemias [57,99].

Metabolic reprogramming has been shown to be associated with the development of certain autoimmune diseases [21]. Pro-inflammatory responses commonly seen in autoimmune diseases are directly associated with a metabolic shift in immune cells to a Warburg-like metabolism [100]. In this line, dimethyl fumarate, which is an immunomodulatory drug used in the treatment of MS among other autoimmune diseases, downregulates aerobic glycolysis in activated immune cells switching their phenotype to a more anti-inflammatory status [101]. Thus, the regulation of immunometabolism is essential in the development and progression of these pathological conditions, and sirtuins, as modulators of cellular metabolism, could play important roles on this. Indeed, some studies have shown that sirtuins are involved in systemic lupus erythematosus and MS [99,102], although the precise mechanism by which they regulate the progression of these diseases is not fully understood. 

A potential mechanism involved in the metabolic regulation of autoimmune diseases by sirtuins could be by controlling the differentiation and activity of immune cells. M1 macrophages have been associated with the increased inflammation observed in many autoimmune diseases such as rheumatoid arthritis [103], and as discussed above, sirtuins are key regulators of macrophage metabolism, which is closely connected with their polarization status (Figure 1). Furthermore, sirtuins regulate the activity of important transcription factors involved in both immune cells activation and metabolism. SIRT1, SIRT2, and SIRT6 inhibit NF-κB signaling in various immune cells [27,33,34,71], which induce glucose uptake and glycolysis during inflammation [67,68]. In addition, SIRT1 suppresses the inflammatory phenotype of macrophages and downregulates the production of pro-inflammatory cytokines through NF-κB deacetylation [104], while downregulation of NF-κB by SIRT2 reduces disease severity in a mouse model of arthritis [33]. 

Besides macrophages, SIRT1 has been shown to inhibit T cell activation, and its deficiency results in the breakdown of T cell tolerance, thus promoting autoimmune diseases [46]. Th17 cells are key in the pathogenicity of many inflammatory and autoimmune diseases [105], and this specific subtype of T cells is known to rely on glycolysis and to express high levels of GLUT1 to support their inflammatory function [84,100]. SIRT1 has been shown to be involved in Th17 differentiation, although whether it functions by promoting or protecting against an inflammatory phenotype is not clear [48,49,50]. On the other hand, Treg cells preferably utilize OXPHOS as energy source and exhibit an anti-inflammatory phenotype [84]. Due to its capacity to maintain self-tolerance and restore immune homeostasis, Treg cells have emerged as possible targets for novel therapies against autoimmune diseases [106]. Of note, SIRT1 deficiency promotes the stability of Treg cells and support their functions, while loss of SIRT3 is associated with impaired Treg cells activity [55,89,90]. However, further research is needed to describe the role of sirtuins in regulating metabolism in these cells and their involvement in autoimmune diseases. 

Metabolic rewiring is a hallmark of cancer cells, which reprogram core cellular metabolic pathways to fulfill the energetic and anabolic needs associated with cell proliferation, growth, and spreading [107]. Tumor-infiltrating immune cells rely on specific metabolic pathways for their activity; therefore, they compete with cancer cells for nutrient availability within the tumor microenvironment (TME). Typically, this metabolic competition results in metabolic stress and impaired antitumor immune responses [108]. While the role of sirtuins on cancer metabolism has been extensively studied [109,110], very little is known about their involvement in the metabolic control of tumor-infiltrating immune cells and its impact on tumor progression. Tumor-associated macrophages (TAMs) are one of the main immune cell types within the TME, and although they are usually considered close to an M2-like phenotype, they are highly glycolytic [111]. A recent study demonstrated that FoxO1 regulates anti-tumor effects in TAMs and its deficiency decreases the glycolytic activity of these macrophages [75]. Intriguingly, SIRT1 and SIRT3 have been shown to inhibit FoxO activity, suggesting that they could be involved in the modulation of TAMs metabolism by inactivating this transcription factor [32,37]. 

Myeloid-derived suppressor cells (MDSCs) are immunosuppressive tumor-infiltrating cells that are responsible of immune cell tolerance. It has been reported that SIRT1 deficiency in MDSCs induces their phenotypic change to M1 macrophages via the mTOR-HIF1α glycolytic pathway, decreasing their suppressive function, promoting inflammation and delaying tumor growth in mice [112]. However, whether other sirtuins regulating this axis could exert similar phenotypical changes in MDSCs impacting tumor growth is not currently known. Treg cells also exhibit immunosuppressive functions and have been associated with a poor prognosis in some types of cancer. Based on this, depletion of Treg cells has been proposed as a novel immunotherapy strategy against cancer [113]. As discussed above, SIRT1 and SIRT3 are involved in the control of Treg cells metabolism and differentiation [55,89,90]. However, whether these sirtuins play a role on tumor progression by regulating the number and activity of tumor-infiltrating Treg cells remains to be elucidated. 

## 4. Concluding Remarks

Cellular metabolism has recently emerged as a key regulator of immune responses by controlling the differentiation, activation, and effector functions of immune cells. In this context, it is not surprising that sirtuins evolved as key molecules linking metabolism to the phenotype and function of immune cells. Due to their reliance on NAD^+^, this family of protein deacylases sense the metabolic status of the cell and coordinate appropriate cellular responses by regulating the epigenetic landscape and the activity of several transcription factors, many of which are well-known regulators of cellular metabolism and immune responses. Indeed, work undertaken during the last decade has established a connection between sirtuin biology and immune responses, but further work will be required to fully understand the sirtuin-dependent metabolic control of immune cell phenotype and function in health and disease. While the role of some sirtuins on immune cells has been clearly demonstrated, evidence is lacking on whether it involves the regulation of metabolism. Additionally, vice versa, in some cases, the involvement of a particular sirtuin in the reprogramming of a specific metabolic pathway is well documented, yet experimental evidence is missing on whether that sirtuin regulates the same metabolic pathway on immune cells. Furthermore, several sirtuins have been described to participate in the control of a specific immune cell type, sometimes with opposite functional outcomes, and thus, more work will be required to elucidate the functional interplay of these proteins in the regulation of the immune system. Nevertheless, based on the data summarized and discussed in this review, we believe that future work will likely shed light on some of these questions, increasing our knowledge on the crossroad of sirtuins and immunometabolism during homeostasis and immune-associated diseases. 

## Figures and Tables

**Figure 1 genes-12-01698-f001:**
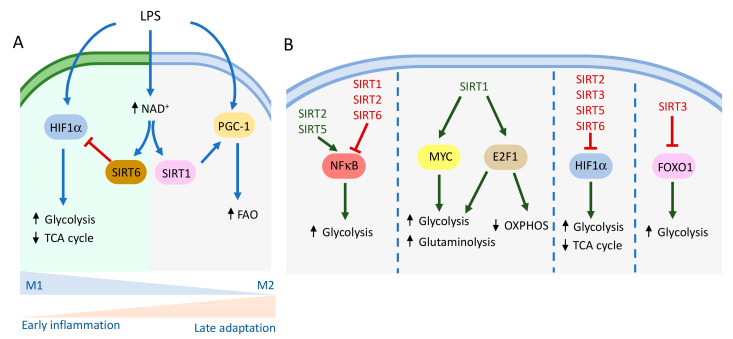
Metabolic control of macrophage fate by sirtuins. (**A**) Proinflammatory conditions lead to an increase in NAD^+^ and SIRT1 and SIRT6 activation, which drive a metabolic reprogramming from glycolysis to fatty acid oxidation during the resolution of the inflammation. (**B**) Sirtuins control macrophage metabolism by regulating the activity of several transcription factors. Blue/green arrows indicate positive regulation, red arrows indicate negative regulation. Up and down black arrows indicate upregulation and downregulation of indicated metabolic pathways, respectively.

**Figure 2 genes-12-01698-f002:**
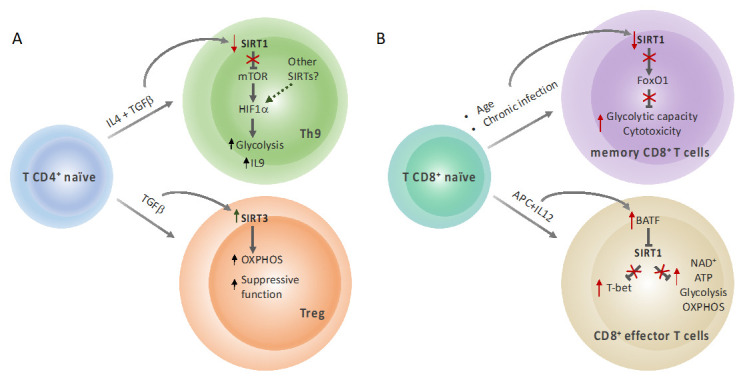
Sirtuins link T cell metabolism and differentiation. (**A**) Differentiation of T CD4+ naïve cell (in blue) into Th9 cells (in green) is accompanied by a decrease in SIRT1 expression, releasing its inhibitory action on the mTOR/HIF1α axis and leading to an increase in aerobic glycolysis and IL9 expression. Differentiation into Treg cells (in orange) is associated with an increase in SIRT3 expression and oxidative phosphorylation, which supports their suppressive function. (**B**) Accumulation of resting memory CD8+ T cells (in violet) during aging or chronic infection is linked to a decrease in SIRT1 expression and the concomitant upregulation of glycolysis. Epigenetic silencing of SIRT1 expression by BATF also drives metabolic changes associated with effector CD8+ T cell differentiation (in beige). Red arrows indicate upregulation (up arrows) or downregulation (down arrows) induced by indicated stimuli. Red “X” indicate block of the depicted pathway regulated by sirtuins.

**Table 1 genes-12-01698-t001:** Summary of main functions and metabolic pathways regulated by sirtuins in immune cells.

Immune Cell	Sirtuin	Known Function	Mechanism of Action	References
Macrophages	SIRT1	Increases FAO	Activation of PGC1α	[25]
		Regulates M2 macrophage polarization	Unknown, probably by increasing oxidative metabolism	[26]
		Enforces oxidative metabolism	Inhibition of NF-κB signaling Activation of AMPK, PPARα and PGC1α	[27,28,29,30]
		Repression of pro-inflammatory cytokine secretion	Inhibition of NF-κB transcriptional activity	[31]
		Regulates insulin levels and glucose metabolism	Inhibition of NF-κB transcriptional activity	[31]
		Regulates macrophage self-renewal	Activation of Myc and E2F1 and repression of FoxO	[32]
	SIRT2	Inhibits pro-inflammatory gene expression	Deacetylation of NF-κB p65	[33,34]
		Induces pro-inflammatory gene expression	Activation of NF-κB	[35]
		Inhibition of FAO and induction of glycolysis (together with SIRT3)	Induce the expression of HIF1α	[36]
	SIRT3	Regulation of M2 macrophage polarization	Blocking the translocation of FoxO1 to the nucleus	[37]
	SIRT5	Promotes the inflammatory response	Promoting the acetylation of NF-κB	[38]
		Suppresses inflammation Inhibition of glycolysis?	Desuccinylation of PKM2	[39]
	SIRT6	Inhibits glucose metabolism	Corepressing HIF1α Inhibiting NF-κB?	[25]
Dendritic cells	SIRT1	Promotes Th2 responses Regulation of glucose metabolism, fatty acid storage and lipid metabolism?	Inhibition of PPARγ	[40,41]
		Promotes Th17 differentiation	Deacetylation of IRF1 and inhibition of IL-27 and IFN-β expression	[42]
		Modulates IL-12 and TFG-1 secretion and the balance of Th1/Treg cells	Inhibition of HIF1α expression	[43]
	SIRT2	Regulates immune response to Salmonella	Induction of NF-κB translocation to the nucleus and NOS2 expression	[44]
	SIRT6	Induces DC migration	Induction of TNF secretion	[45]
T cells	SIRT1	Reduces the production of Th1 and Th2 cytokines	Inhibition of AP-1	[46]
		Represses glycolytic metabolism and Th9 differentiation	Inhibition of mTOR-HIF1α axis	[47]
		Promotes Th17 differentiation	RORγt deacetylation	[48]
		Suppresses Th1 and Th17 differentiation	STAT3 deacetylation	[49,50]
		Inhibits Treg suppressive function	Inhibition of *FoxP3* expression Deacetylation of p53	[51,52]
		Inhibits CD8+ memory T cell differentiation	Activation of FoxO1 and inhibition of glycolytic metabolism	[53]
		Blocks CD8+ effector T cell differentiation	Epigenetic repression of T-bet and inhibition of NAD^+^ and ATP production	[54]
	SIRT3	Promotes Treg suppressive function	Induction of oxidative metabolism	[55]
B cells	SIRT1	Suppresses lymphotoxin and TNF-α production	Inhibition of NF-κB?	[56]
		Induces proliferation of CLL cells	Unknown	[57]
	SIRT2	Induces proliferation of CLL cells	Unknown	[57]
	SIRT3	Suppresses CLL	Deacetylation of IDH2 and SOD2 and inhibition of glycolysis	[58]

## Data Availability

Not applicable.

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
