# Peer review of "Sirtuins as Metabolic Regulators of Immune Cells Phenotype and Function"

_genes, 2021, doi:10.3390/genes12111698_

Round 1
Reviewer 1 Report
In their review « Sirtuins as metabolic regulators of immune cells phenotype and function” Fortuny and Sebastian, provide a summary of the literature on the roles of Sirtuin proteins in the regulation of some immune cell functions and their implications in diseases. This is an original review and I am sure that it will be of interest for the readers of Genes and beyond.
The review is properly written but some aspects need to be better explain. I have recommendations that may help to further improve clarity
General comments :
- It will be informative and clearer for the readers if the authors could specify when it is appropriate In which mouse models the results where obtain (global KO, conditional KO, steady state or not).
- Please add a table which summarize the tissue and the cell distribution of sirtuins with known or suppose effect on their phenotype/function/metabolism.
- What is it known about sirtuins and other immune cells such as NK cells, NK-T cells, neutrophils… ? Please add some words about it
- What is it known about the role/behavior of sirtuins in metabolic disorders ?
- What about the role of sirtuins in human immune cells and in human diseases ?
Specific comments:
Page 2, line 87 : the introducing paragraph is not clear, please rephrase and provide a better definition of innate (and adaptive immunity). Indeed, it is now well admitted that the behavior of almost all innate immune cells influence adaptive immunity. The dichotomy propose by the authors is not appropriate.
Page 4, line 173/174 : please specify which phenotype are “activated macrophages” : M1 or M2 ?
Page 8, line 371: please specify what are the molecules used and how they have been used
Page 12, line 557 ; this paragraph repeats almost what was mentioned page 10. Please rephrase and/or remove.
Minor points :
Page 1, line 26 : end of the line, remove “is” after SIRT6
Page 3, line 118: change “or” by “of” after expression.
Author Response
RESPONSE TO THE REVIEWERS
We would like to thank the reviewers for their insightful comments and highly constructive suggestions, which have helped us to significantly improve our manuscript. Both reviewers agreed on the interest of the topic covered and appreciated our efforts to summarize the current knowledge about the role of sirtuins on regulating metabolism in immune cells. However, they raised several concerns that we hope to have satisfactory addressed, as detailed point-by-point below. In short, we have edited the text to make clearer some statements, corrected grammar errors and misspellings and added a new table (Table 1) summarizing the functional contribution of sirtuins to the control of immune cells phenotype and metabolism. We hope the reviewers will find now the manuscript suitable for publication.
Reviewer#1:
In their review « Sirtuins as metabolic regulators of immune cells phenotype and function” Fortuny and Sebastian, provide a summary of the literature on the roles of Sirtuin proteins in the regulation of some immune cell functions and their implications in diseases. This is an original review and I am sure that it will be of interest for the readers of Genes and beyond.The review is properly written but some aspects need to be better explain. I have recommendations that may help to further improve clarity
We thank this reviewer for the overall appreciation of our manuscript and considering it of interest for the scientific community.
General comments:
- It will be informative and clearer for the readers if the authors could specify when it is appropriate In which mouse models the results where obtain (global KO, conditional KO, steady state or not).
We agree with this reviewer on the relevance of the specific mouse model employed in the studies summarized in our manuscript and we apologize for this omission. We have now specified this in the revised version of our manuscript.
- Please add a table which summarize the tissue and the cell distribution of sirtuins with known or suppose effect on their phenotype/function/metabolism.
In our original manuscript, we included two figures to summarize the main roles of sirtuins on the regulation of metabolism in macrophages and T cells. However, we agree with this reviewer that adding a table including other immune cells covered in our manuscript with a more detailed information (a concern also raised by Reviewer#2) will be helpful for the reader. We have included this information in the new Table 1 of our revised manuscript.
- What is it known about sirtuins and other immune cells such as NK cells, NK-T cells, neutrophils… ? Please add some words about it
We thank the reviewer for bringing this up. Indeed, to our knowledge, there is very little known about the role of sirtuins on NK cells, NKT cells and neutrophils with regards to metabolism. We have included a paragraph stating this at the end of section 2.1 that reads as follows: “Besides macrophages and DCs, the role of sirtuins on the metabolic control of other innate immune cells has not been described. However, it is known that neutrophils undergo a metabolic shift towards glycolysis to support their phagocytic activity [68]. Based on this, one can hypothesize that those sirtuins involved in the control of glycolysis could regulate this metabolic pathway in neutrophils as well.”
- What is it known about the role/behavior of sirtuins in metabolic disorders?
We thank this reviewer for his/her comment. Much is known about the implication of several sirtuins in metabolic disorders, such as cardiovascular diseases, obesity, metabolic syndrome, etc. Excellent reviews from the Auwerx and Guarente laboratories, for instance, have covered these topics. In our manuscript, we focused on the role of this family of proteins on the metabolic control of immune cell fate and phenotype, an emerging and less covered topic in the current scientific literature. We hope that this reviewer will agree with us that including a section about metabolic disorders would be out of the scope of our manuscript and would make it unnecessarily extensive.
- What about the role of sirtuins in human immune cells and in human diseases?
Most of the studies included in our manuscript have been done in mouse models. However, whether the same phenotypes/functions are conserved in humans remains to be elucidated. There are few cases, though, where this information is available, and we have included few sentences in our revised manuscript to highlight it (see for instance line 402, 522 and 555).
Specific comments
Page 2, line 87 : the introducing paragraph is not clear, please rephrase and provide a better definition of innate (and adaptive immunity). Indeed, it is now well admitted that the behavior of almost all innate immune cells influence adaptive immunity. The dichotomy propose by the authors is not appropriate.
We agree with this reviewer on the lack of clarity of this paragraph. We have removed the adaptive-innate dichotomy from this paragraph and better defined both innate and adaptive immunity in separate paragraphs at the beginning of both sections (new lines 102-106 and 334-338).
Page 4, line 173/174 : please specify which phenotype are “activated macrophages” : M1 or M2 ?
We have added this information (new line 198).
Page 8, line 371: please specify what are the molecules used and how they have been used
We have specified the activators and inhibitors of SIRT1 in this paragraph (new lines 426-430).
Page 12, line 557 ; this paragraph repeats almost what was mentioned page 10. Please rephrase and/or remove.
We have removed this paragraph.
Minor points:
Page 1, line 26 : end of the line, remove “is” after SIRT6
Corrected.
Page 3, line 118: change “or” by “of” after expression.
Corrected.
Reviewer 2 Report
In the manuscript entitled “Sirtuins as metabolic regulators of immune cells phenotype and function” the authors describe metabolic changes in the innate and adaptive immune response. They are focusing on the role of members of the Sirtuin-family in these metabolic processes and changes.
The authors focus more on describing the different metabolic changes than discussing functional involvements of sirtuins. A description of how sirtuin-mediated epigenetic changes induce the described metabolic changes on a functional level is missing and the major weakness of this review.
The manuscript is difficult to read due to the structure and insufficient use of the English language. It will help the reader if the reviewed data is presented in additional tables for innate immune cells and adaptive immune cells. A table describing each sirtuin and its involvement in the immune response would be helpful too.
- line 26&27: SIRT6 is and SIRT 7 have been – please correct the grammar in this sentence.
- line 27&28: Please clarify the subcellular localization of the SIRT proteins. How do they differ from the subcellular localization stated in the sentence: “SIRT1 and SIRT2 can be found in the cytoplasm 25 and in the nucleus; SIRT3, SIRT4 and SIRT5 are mitochondrial sirtuins while SIRT6 is and 26 SIRT7 have been reported to be found mainly in the nucleus.”
- line 63 explain abbreviation OXPHOS.
- line 71: What is the meaning of synthetic precursors here?
- line 94-96: Meaning of these sentences is unclear: “Macrophages are specialized immune cells involved in the phagocytosis and elimination of pathogens and death or aberrant cells. They are also responsible for antigen presentation to T cells leading to the production of inflammatory cytokines, thus coordinating and effective immune response.”
- line 102&103: Define M1 and M2.
- lines 112 to 118: These paragraph is difficult to understand. What is related to M1 and what is related to M1? The use of “On the contrary” and “In contrast” seems illogical.
- line 163: empty space
- line 163-165: Please correct sentence: “NF-B has been shown to induce glycolysis by directly increasing the expression of the glucose transporter GLUT3 [32] and upregulates glycolytic metabolism during acute inflammation [33].”
- line 292-294: Sentence is difficult to read and to understand: “The adaptive immune response is a complex response that leads to the generation of specific antibodies and cellular responses that allow the efficient elimination of the antigen and the production of memory cells.”
- line 360: empty space
- line 580: Please correct the sentence: “And vice versa, in some cases where it is well documented the involvement of a particular sirtuin in the reprogramming of a specific metabolic pathway (shown to regulate immune cell function), experimental evidence is missing on whether that sirtuin regulates that metabolic pathway on immune cells.”
Author Response
RESPONSE TO THE REVIEWERS
We would like to thank the reviewers for their insightful comments and highly constructive suggestions, which have helped us to significantly improve our manuscript. Both reviewers agreed on the interest of the topic covered and appreciated our efforts to summarize the current knowledge about the role of sirtuins on regulating metabolism in immune cells. However, they raised several concerns that we hope to have satisfactory addressed, as detailed point-by-point below. In short, we have edited the text to make clearer some statements, corrected grammar errors and misspellings and added a new table (Table 1) summarizing the functional contribution of sirtuins to the control of immune cells phenotype and metabolism. We hope the reviewers will find now the manuscript suitable for publication.
Reviewer#2:
In the manuscript entitled “Sirtuins as metabolic regulators of immune cells phenotype and function” the authors describe metabolic changes in the innate and adaptive immune response. They are focusing on the role of members of the Sirtuin-family in these metabolic processes and changes.
The authors focus more on describing the different metabolic changes than discussing functional involvements of sirtuins. A description of how sirtuin-mediated epigenetic changes induce the described metabolic changes on a functional level is missing and the major weakness of this review.
In our manuscript, we first provide an overview of the main metabolic pathways regulating immune cells fate and function. Then, we integrate this information with known roles of sirtuins on the control of these metabolic pathways on immune cells and, when described in the original studies, we summarize the main mechanisms involved. We agree with this reviewer that an important function of some sirtuins is to link the cellular metabolic status with epigenetic changes driving gene expression. However, little is known on how sirtuins epigenetically regulate cellular metabolic pathways specifically in immune cells. In our revised manuscript, we describe now how SIRT6 regulates glycolysis by deacetylating H3K9 and H3K56 on the promoters of glycolytic genes in macrophages, which, together with SIRT1-dependent epigenetic regulation of T-bet expression during the differentiation of CD8 memory T cells, provide two of the very few demonstrations of the crosstalk between sirtuin-mediated epigenetics and metabolism in immune cells.
The manuscript is difficult to read due to the structure and insufficient use of the English language. It will help the reader if the reviewed data is presented in additional tables for innate immune cells and adaptive immune cells. A table describing each sirtuin and its involvement in the immune response would be helpful too.
We thank the reviewer for this comment, which has helped us to improve the writing and to correct several typos present in the original manuscript. We have also included a table following this reviewer’s suggestion (see also response to reviewer#1).
- line 26&27: SIRT6 is and SIRT 7 have been – please correct the grammar in this sentence.
We apologize for this mistake, which we have corrected in the revised manuscript.
- line 27&28: Please clarify the subcellular localization of the SIRT proteins. How do they differ from the subcellular localization stated in the sentence: “SIRT1 and SIRT2 can be found in the cytoplasm 25 and in the nucleus; SIRT3, SIRT4 and SIRT5 are mitochondrial sirtuins while SIRT6 is and 26 SIRT7 have been reported to be found mainly in the nucleus.”
We have included few sentences in this paragraph to clarify this point (new lines 29-34).
- line 63 explain abbreviation OXPHOS.
Corrected.
- line 71: What is the meaning of synthetic precursors here?
We have rephrased it for clarity and now it reads “biosynthetic precursors”, in a clear reference to anaplerotic metabolites required to build up macromolecules.
- line 94-96: Meaning of these sentences is unclear: “Macrophages are specialized immune cells involved in the phagocytosis and elimination of pathogens and death or aberrant cells. They are also responsible for antigen presentation to T cells leading to the production of inflammatory cytokines, thus coordinating and effective immune response.”
We have rephrased this sentence (new lines 108-111).
- line 102&103: Define M1 and M2.
For clarity, we have added “pro-inflammatory M1 macrophages” and “anti-inflammatory M2 macrophages”.
- lines 112 to 118: These paragraph is difficult to understand. What is related to M1 and what is related to M1? The use of “On the contrary” and “In contrast” seems illogical.
We kindly disagree with this reviewer on this. The paragraph this reviewer is referring to reads as follows: “Pro-inflammatory M1 macrophages mainly rely on glycolysis through the activation of several transcription factors such as HIF1-α and exhibit an impaired tricarboxylic acid (TCA) cycle. On the contrary, anti-inflammatory M2 macrophages maintain an active TCA cycle to support OXPHOS for energy production.” This paragraph describes the main metabolic adaptations of both M1 and M2 macrophages. While M1 macrophages are more glycolytic and inhibit oxphos, M2 macrophages display a metabolically opposite phenotype with an active oxphos and decreased glycolysis. Therefore, in our opinion, the use of “on the contrary” does not seem illogical as it highlights opposite phenotypes.
- line 163: empty space
Corrected.
- line 163-165: Please correct sentence: “NF-B has been shown to induce glycolysis by directly increasing the expression of the glucose transporter GLUT3 [32] and upregulates glycolytic metabolism during acute inflammation [33].”
Corrected (new line 189)
- line 292-294: Sentence is difficult to read and to understand: “The adaptive immune response is a complex response that leads to the generation of specific antibodies and cellular responses that allow the efficient elimination of the antigen and the production of memory cells.”
We agree with this reviewer that this sentence is difficult to read. We have rephrased it (new lines 334-337)
- line 360: empty space
Corrected.
- line 580: Please correct the sentence: “And vice versa, in some cases where it is well documented the involvement of a particular sirtuin in the reprogramming of a specific metabolic pathway (shown to regulate immune cell function), experimental evidence is missing on whether that sirtuin regulates that metabolic pathway on immune cells.”
We have corrected this sentence to better convey the main message. It now reads: “And vice versa, in some cases is well documented the involvement of a particular sirtuin in the reprogramming of a specific metabolic pathway, yet experimental evidence is missing on whether that sirtuin regulates the same metabolic pathway on immune cells.”